# Morphology and mobility as tools to control and unprecedentedly enhance X-ray sensitivity in organic thin-films

Inés Temiño[1,2,5], Laura Basiricò [3,4,5], Ilaria Fratelli[3,4], Adrián Tamayo[1,2], Andrea Ciavatti [3,4], Marta Mas-Torrent[1,2]✉ & Beatrice Fraboni [3,4]✉

Organic semiconductor materials exhibit a great potential for the realization of large-area solution-processed devices able to directly detect high-energy radiation. However, only few works investigated on the mechanism of ionizing radiation detection in this class of materials, so far. In this work we investigate the physical processes behind X-ray photoconversion employing bis-(triisopropylsilylethynyl)-pentacene thin-films deposited by bar-assisted meniscus shearing. The thin film coating speed and the use of bis-(triisopropylsilylethy-nyl)-pentacene:polystyrene blends are explored as tools to control and enhance the detection capability of the devices, by tuning the thin-film morphology and the carrier mobility. The so-obtained detectors reach a record sensitivity of $1.3 \cdot 10^4$ μC/Gy·cm², the highest value reported for organic-based direct X-ray detectors and a very low minimum detectable dose rate of 35 μGy/s. Thus, the employment of organic large-area direct detectors for X-ray radiation in real-life applications can be foreseen.

[1] Institut de Ciència de Materials de Barcelona (ICMAB-CSIC), Campus de la UAB, 08193 Bellaterra, Spain. [2] CIBER-BBN, Campus de la UAB, 08193 Bellaterra, Spain. [3] Department of Physics and Astronomy, University of Bologna, Viale Berti Pichat 6/2, 40127 Bologna, Italy. [4] National Institute for Nuclear Physics—INFN section of Bologna, Bologna, Italy. [5] These authors contributed equally: Inés Temiño, Laura Basiricò. ✉email: mmas@icmab.es; beatrice.fraboni@unibo.it

The employment of organic semiconductors as active material for highly sensitive ionizing radiation detection systems has been assessed in the last 10 years by several studies reported in the literature[1–12]. The efforts spent by the scientific research community to conceive and realize innovative X- and gamma-ray detectors based on organic molecules and polymers originate from the exceptional properties of these materials, above all the possibility to deposit them from solution onto nonconventional substrates and over large areas by means of easy low-cost processing techniques[13,14].

Another relevant feature of organic materials is their low atomic number Z. On one hand, this offers the unique characteristic of being human tissue equivalent, which is a crucial requirement for medical dosimetry applications. However, on the other hand, it represents a constraint for the detection of high-energy radiation due to the resulting low stopping power. A wide range of strategies were recently suggested to enhance the absorbance of the organic layer, such as introducing in the active layer high-Z elements as nanoparticles[2] or quantum dots[5], coupling the added high-Z elements with organic bulk heterojunctions[5,6], inserting carbon nanotubes in the active layer[10], and, more recently, by adding high-Z atoms directly into the molecule structure[15].

Such peculiarities of organic materials open the way for several innovative applications, spanning from medical diagnostics to public safety, space, cultural heritage, and environmental monitoring, since large-area, light-weight, low-cost, and flexible devices could overcome the limitations of traditional inorganic semiconductor X-ray detectors (e.g., amorphous-Si, amorphous-Se, poly-CZT, diamond)[16–20].

However, despite the notable number of recent papers pushing the performance of organic materials for the direct X-ray detection, only a few of them investigated the physical processes ruling the detection mechanism of high-energy photons in such low-Z molecular systems[7,21]. In particular, in our previous work[21] we attributed the direct X-ray photoconversion observed in microcrystalline thin films of bis-(triisopropylsilylethynyl) pentacene (TIPS-pentacene) to an electron traps assisted photoconductive gain mechanism, where the magnitude and the dynamics of the response together with the high sensitivity to the X-rays of the organic thin-film devices could be rationalized, thanks to a developed kinetic model.

In this work we target to deeply step into the understanding of the physical processes and the parameters controlling the direct detection of ionizing radiation in organic thin films. To this aim, we employed TIPS-pentacene-based organic field-effect transistors (OFETs) fabricated on Si/SiO$_2$ substrates by a large-area and scalable deposition technique, namely bar-assisted meniscus shearing (BAMS)[22], as direct X-ray detectors. BAMS technique has been demonstrated as an effective low-cost and fast speed fabrication process leading to organic semiconducting thin films with high and reproducible field-effect mobility values[23] and with a controlled thin-film morphology[24]. Moreover, the combination of BAMS with the blending of the organic semiconducting small molecules with an insulating polymer (such as polystyrene (PS)) has shown to enhance thin-film processability and homogeneity[25–27]. In this work we tune the carrier mobility of the OFETs by controlling the TIPS-pentacene:PS blending ratio, and the grain size and grain boundaries density via regulation of the coating speed. With this powerful tool we succeed not only in tuning and enhancing the sensitivity of the detectors but also in studying the role of the electrically active defects related to the grain boundaries in the detection process. We demonstrate that carrier mobility and thin-film morphology (i.e., number and size of grain boundaries) are the two main parameters affecting the photoconductive gain process and the carrier trapping effects,

that is, the physical processes that explain why thin organic films can directly detect high-energy radiation. We reduce hole traps by passivating the hydroxyl groups of SiO$_2$ with PS and we tune electron traps by controlling the grain boundaries density. By selectively controlling both factors, we achieve a very low minimum detectable dose rate of 35 $\mu$Gy s$^{-1}$ and a record sensitivity of $1.3 \cdot 10^4$ $\mu$C Gy$^{-1}$ cm$^{-2}$, the highest reported for organic-based detectors[6,15] and three orders of magnitude higher than the one of a-Se[28] (i.e., the inorganic semiconductor presently used for large-area direct radiation detectors). Thanks to the uniformity and reliability of the employed fabrication process, we implemented a proof-of-principle X-ray sensor based on Wheatstone bridge geometry to further control and stabilize the response.

## Results

**BAMS-coated TIPS-pentacene OFETs as X-ray detectors.** Bottom-gate bottom-contact OFETs based on TIPS-pentacene were fabricated by solution depositing TIPS-pentacene thin films ($\approx$80 nm thick) by BAMS technique on Si/SiO$_2$ substrates with interdigitated gold electrodes (the cross section is depicted in Fig. 1a) at a speed of 10 mm/s. The polarized optical microscope image in Fig. 1b shows that BAMS technique provides polycrystalline thin films with high coverage, homogeneity, crystallinity, and spherulitic domains. Devices were electrically characterized under ambient conditions, with $V \geq -20$ V for both $V_{SD}$ and $V_{SG}$. Typical OFET transfer and output characteristics are reported in Fig. 1c and Supplementary Fig. 1, respectively, showing excellent electrical features such as good current modulation and negligible contact resistance. Average values of the main OFET parameters were extracted in the saturation regime for the fabricated devices and are collected in Supplementary Table 1. Charge carrier mobility values are about $10^{-2}$ cm$^2$ V$^{-1}$ s$^{-1}$, while the mean threshold voltage is $V_{th} = 6 \pm 2$ V indicating that the devices are slightly doped with an excess of positive charges. The on/off ratio ($I_{on}/I_{off}$) is around $10^3$, related to a not so steep switch-on of the devices (average subthreshold swing SS = $1.6 \pm 0.4$ V dec$^{-1}$).

The devices were characterized as X-ray detectors, measuring the induced photocurrent upon multiple on/off beam switching cycles. The employed setup is shown in Fig. 1d, more details on the experimental conditions are reported in Methods. Figure 1e shows the real-time response of a BAMS-coated TIPS-pentacene device in saturation regime ($V_{SD} = -20$ V, $V_{SG} = -15$ V) exposed to a Mo-target X-ray tube with a dose rate of 53 mGy s$^{-1}$. As previously reported[21,29], the observed photocurrent response can be ascribed to a photoconductive gain mechanism related to electron traps as detailed below. The X-ray induced photocurrent (evaluated as the amplitude of the on peaks with respect to the current base line) as a function of the dose rate of the incident radiation is shown in Supplementary Fig. 2. From the slope of this plot a high sensitivity per unit area of $3.8 \cdot 10^2$ $\mu$C Gy$^{-1}$ cm$^{-2}$ was estimated (Table 1), about one order of magnitude higher than previously reported results on drop casted TIPS-pentacene OFET based detectors[29]. All the sensitivity values reported hereafter have been calculated considering the photocurrent values after 60 s of exposure to X-rays. In fact, due to the transient nature of the X-ray induced photocurrent, the sensitivity increases with time exposure, reaching saturation after about 50 s (Supplementary Fig. 3).

Indeed, the radiation response offered by this class of direct organic detectors is described by the photoconductive gain mechanism that amplifies the X-ray photogenerated current by a factor G[21,30,31]. This process of amplification is activated by the trapping of minority charge carriers and the factor G can be expressed as the ratio between the recombination time ($\tau_r$) and

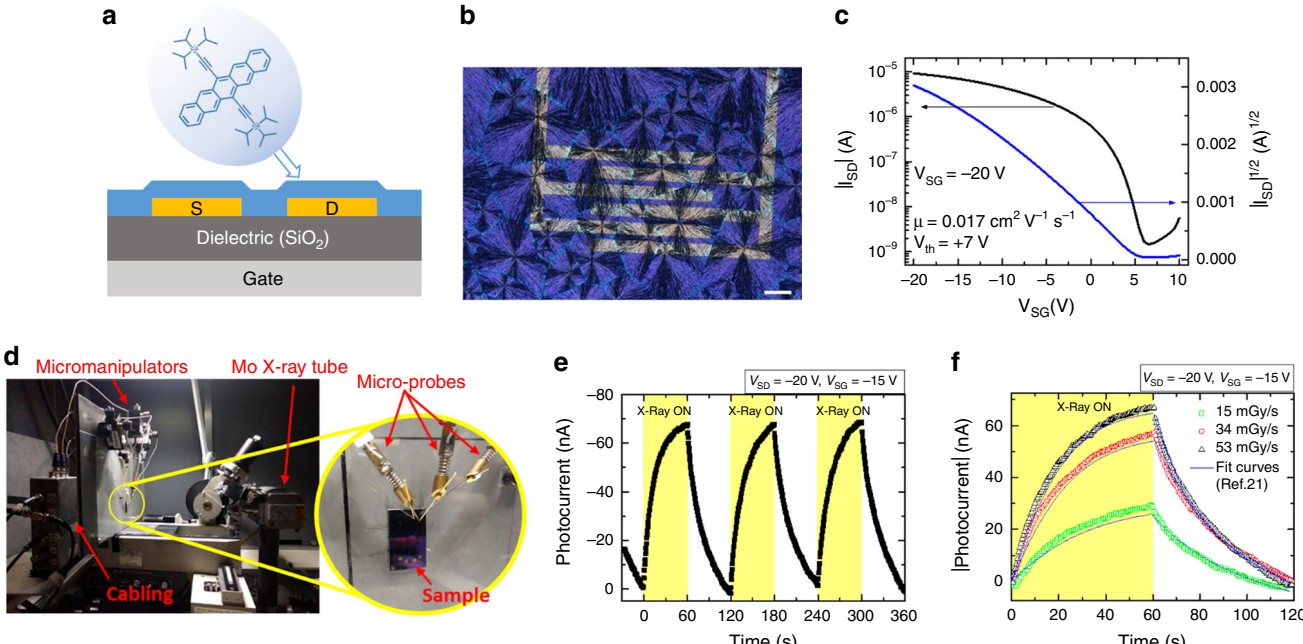

**Fig. 1 BAMS deposited TIPS-pentacene OFETs as X-ray detectors. a** Schematic representation of the bottom-gate bottom-contact OFETs employed in this work, showing the cross section of the devices and the molecular structure of TIPS-pentacene. **b** Cross-polarized optical microscope image of a pure TIPS-pentacene thin film deposited by BAMS on a Si/SiO₂ substrate with pre-patterned interdigitated electrodes. Scale bar is 100 μm. **c** Typical transfer characteristics in the saturation regime of a pure TIPS-pentacene OFET. **d** Photograph of the experimental setup for the samples irradiation with a Mo-target X-ray tube. **e** X-ray induced photocurrent response of a pure TIPS-pentacene BAMS-coated device upon three on/off switching cycles (yellow areas, time windows of 60 s) employing a dose rate of 53 mGy s⁻¹, reaching a high sensitivity value of $3.8 \cdot 10^2$ μC Gy⁻¹ cm⁻². **f** Experimental and fitted curves of the response of the same device for three different dose rates of radiation yielding an amplification of the photocurrent of $2 \cdot 10^6$. The fit curves have been obtained by applying the analytical model described in ref. [21].

**Table 1 Structural, electrical, and detection parameters for different deposition speeds.**

| Deposition speed (mm s⁻¹) | Grain size (μm²) | Thickness (nm) | Mobility (cm² V⁻¹ s⁻¹) | $N_T$ (10¹² eV⁻¹ cm⁻²) | Sensitivity (μC Gy⁻¹ cm⁻²) | Gain |
|---|---|---|---|---|---|---|
| Low (4) | 17 ± 3 | 70 ± 20 | $(2.5 \pm 0.7) \cdot 10^{-2}$ | 1.7 ± 0.4 | $(1.0 \pm 0.2) \cdot 10^2$ | $(7 \pm 2) \cdot 10^5$ |
| Standard (10) | 6 ± 2 | 80 ± 20 | $(1.7 \pm 0.5) \cdot 10^{-2}$ | 1.8 ± 0.5 | $(3.8 \pm 0.1) \cdot 10^2$ | $(2.0 \pm 0.6) \cdot 10^6$ |
| High (28) | 6 ± 3 | 120 ± 50 | $(2.4 \pm 0.6) \cdot 10^{-2}$ | 1.6 ± 0.4 | $(3.8 \pm 1.2) \cdot 10^2$ | $(4.0 \pm 1.5) \cdot 10^6$ |

the transit time $(\tau_t)$

$$G = \frac{\tau_r}{\tau_t}. \tag{1}$$

These two characteristic times represent respectively the time of recombination of the minority carriers (electrons in the case of TIPS-pentacene) trapped in the organic layer, i.e., the minority carrier lifetime, and the transit time of the majority carriers (holes in the case of TIPS-pentacene) along the channel of the OFET

$$\tau_t = \frac{L^2}{\mu V}, \tag{2}$$

where $L$ is the channel length, $\mu$ is the holes mobility, and $V$ is the applied bias.

The recombination time can be defined as the following expression:

$$\tau_r(\rho_x) = \frac{\alpha}{\gamma} \left[ \alpha \ln\left(\frac{\rho_0}{\rho_x}\right) \right]^{\frac{1-\gamma}{\gamma}}, \tag{3}$$

where $\alpha$ is the timescale in which the relaxation after the irradiation takes place, $\gamma$ represents the width of the distribution of relaxation timescale $\alpha_i$ (typically $\gamma < 1$ in amorphous and polycrystalline materials), $\rho_0$ is the carrier density in the saturation condition, and $\rho_x$ the carrier density induced by a certain dose of radiation. By fitting the current curve obtained after irradiating the detector at different radiation doses (X-rays switched off), one has the possibility to calculate all the parameters that describe the relaxation time of the detector identified by the stretched exponential in Eq. (3). Thanks to this, it is possible to calculate the amplification factor which determines the gain of the photocurrent directly induced by the absorbed photons.

The dynamic behavior of the photocurrent signal of the here reported organic detectors was excellently fitted by such photoconductive gain model[21] with exponential trap distribution using a single set of fitting parameters for all the different dose rates employed, as shown in Fig. 1f.

The huge amplification of the photocurrent upon exposure to X-rays gives a gain value of $2 \cdot 10^6$ (Table 1). Significantly, the same gain model well described the dynamic of the X-ray response of TIPS-pentacene thin films both employing Si/SiO₂ substrates (the present work), and PET plastic substrates[21,29], providing a further evidence that the model can be generalized to different types of organic materials, substrates, and device geometries.

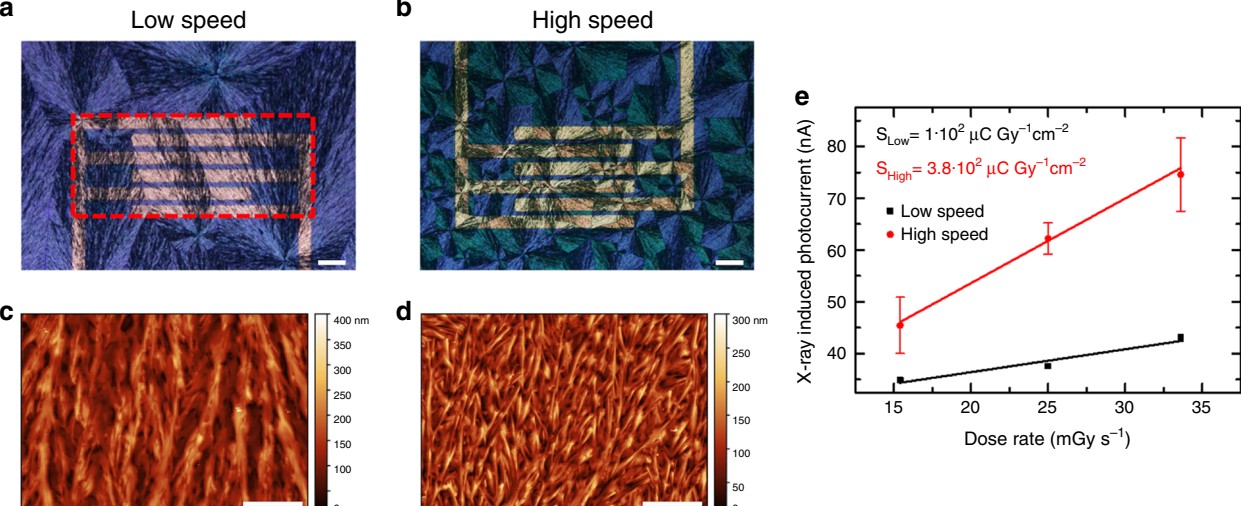

**Fig. 2 Impact of film morphology (grain size/density) on X-ray detection sensitivity.** Cross-polarized optical microscope images and AFM topography images of pristine TIPS-pentacene thin films deposited by BAMS at **a**, **c** low speed (4 mm s$^{-1}$) and **b**, **d** high speed (28 mm s$^{-1}$). The dashed red box in **a** indicates the pixel area used for the calculation of the sensitivity. Scale bar: 100 μm. **e** X-ray induced photocurrent versus dose rate plot for the two types of films and relative calculated sensitivity, obtained under irradiation by a Mo-target X-ray tube. The error bars refer to the statistical fluctuations of the signal amplitude over three on/off switching cycles of the X-ray beam in the same condition.

**Impact of film morphology on X-ray detection sensitivity.** In order to discern the role that the thin-film morphology plays on the X-ray detection mechanism, TIPS-pentacene-based thin films were solution deposited at different coating speeds employing the same patterned silicon substrates. The modification of the deposition speed is known to influence the crystallization regime of the thin film, affecting the morphology, the domain size, and even the molecular orientation[32,33]. For TIPS-pentacene BAMS-coated films, we found that by controlling the coating speed of the semiconductor layer it was possible to tune the dimension of the crystal domains. In particular, coating at a speed of 4–5 mm s$^{-1}$ gives rise to the same spherulitic domains found when coating at a speed of 10–30 mm s$^{-1}$ but with considerably bigger domains size. This can be clearly observed in the polarized optical microscope images of two different samples fabricated at a coating speed of 4 and 28 mm/s, respectively, shown in Figs. 2a, b. While the film thickness does not vary considerably (Supplementary Fig. 4), the atomic force microscopy (AFM) images in Figs. 2c, d show morphological features that indicate bigger size of TIPS-pentacene crystallites for the low-speed processed thin film, as reported in Table 1 (see Supplementary Fig. 5). However, it should be noted that increasing the coating speed from 10 to 28 mm s$^{-1}$ does not significantly affect the thin-film morphology or the grain size, as observed by comparison of the corresponding optical microscope and AFM images (Supplementary Fig. 6). Increasing further the coating speed leads to smaller but also less uniform domain sizes; hence, higher speeds have been avoided in order to achieve reproducible thin films.

As it can be seen in Fig. 2e, the two types of samples exhibit a very different X-ray detection capability: the devices processed at high speed show a sensitivity more than three times higher than that of the low-speed processed devices (3.8 · 10$^2$ and 1 · 10$^2$ μC Gy$^{-1}$ cm$^{-2}$, respectively). Since the mobility and the hole trap density ($N_T$) values (Table 1), extracted from the OFETs characteristics, are comparable, the differences observed in the X-ray response must be ascribed to the different morphologies and, in particular, to the different density of grain boundaries. In fact, the higher photoconductive gain estimated for the high-speed processed device (Table 1) points out that, in these samples,

there is a higher number of electron traps inducing a greater amplification of the photocurrent.

In fact, the presence of the electron traps impacts on the recombination time $\tau_r$, i.e., the electron lifetime, which is directly proportional to the gain (Eq. 1). The correlation existing between the electron traps, the density of grain boundaries determined by the deposition speed, and the sensitivity to the radiation can be then explained comparing the experimentally determined sensitivity and hole mobility with the analytical calculation of $\tau_t$ and $\tau_r$ for different deposition speeds. These calculations have been carried out by fitting the experimental data with the stretched exponential decay of the photocurrent after irradiation, due to the slow relaxation of the trapped charge carriers, foreseen by the photoconductive gain model (Eq. 2). The resulting values are reported in the graphs in Fig. 3: both the recombination time (related to electron traps) and the sensitivity increase with increasing deposition speed, i.e., with the reduction of grain size and, as a consequence, with the increasing of grain boundary density. A plateau is reached for similar grain sizes (standard and high speed). On the other hand, we could assess how the experimentally measured hole mobility and the related hole transit time show a markedly different trend.

These results indicate that an efficient strategy to enhance the X-ray sensitivity is the reduction of the grains size (or an increase in their density) in the active layer, i.e., the enhancement of the electron trap density, which, in this case, is achieved by increasing the coating speed.

Further, from the values reported in Table 1, it can be also inferred that the thickness of the active layer does not have a strong impact on the sensitivity of the here reported X-ray detectors: samples with similar thickness (low and standard) exhibit different sensitivity and gain values. In fact, the here presented detectors are based on a field-effect transistor structure where the charge transport occurs at the interface between the organic semiconductor and the dielectric layer, i.e., the few nanometers of the transistor channel. This differs from what happens in vertical stacked organic detectors, e.g., the one recently reported in the literature by Jayawardena et al.[8], who presented a 100-μm-thick P3HT:PCBM:Bi$_2$O$_3$ heterojunction device, where charge transport occurs through

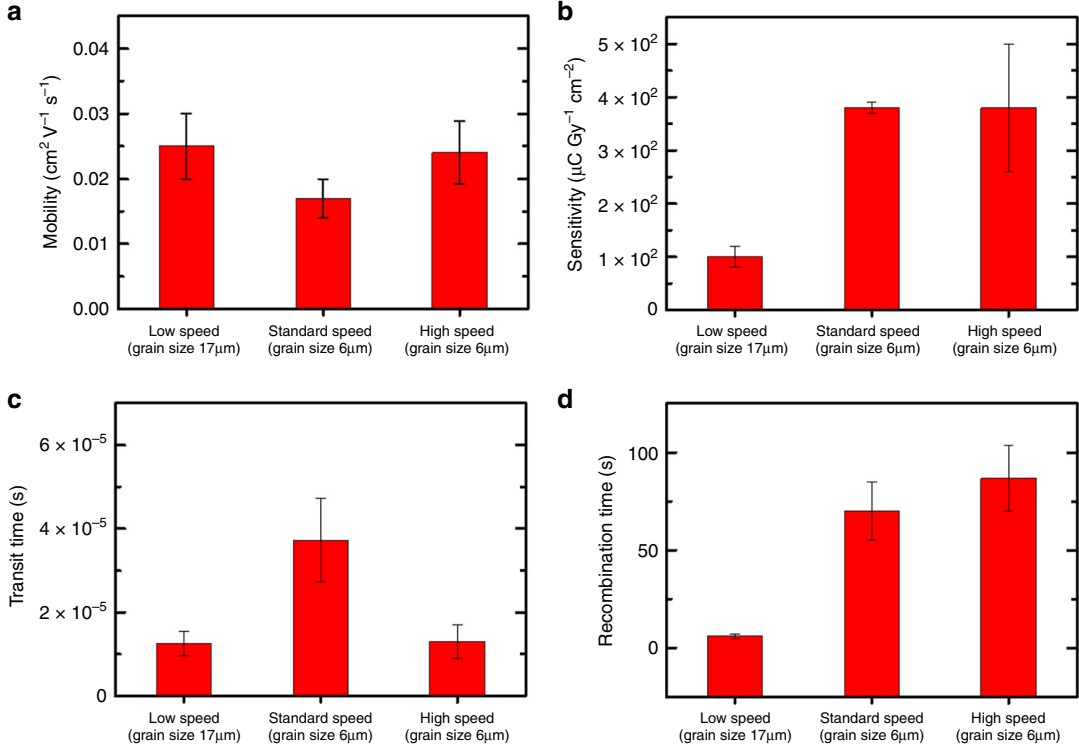

**Fig. 3 Correlation between mobility, sensitivity, and traps.** Experimentally determined hole mobility (**a**) and sensitivity (**b**); analytically determined transit time (**c**) and recombination time (**d**) for pure TIPS-pentacene-based samples deposited at different speeds (i.e., with different grain sizes). The error bars refer to the statistical fluctuations of the parameters over four samples for each deposition speed.

the bulk of the organic layer, and thus its thickness plays a crucial role in the detection process. In our case the radiation-induced charge carriers involved in the photoconductive gain process flow within the OFET channel, meaning that the detection performance is more affected by the efficacy of the transport process at the semiconductor/dielectric interface than by the efficacy of the radiation absorption in the semiconductor bulk. For this reason, increasing the active layer thickness does not represent a winning strategy to improve the X-ray detection properties of the here reported devices.

**Impact of OFET properties on X-ray sensitivity: TIPS-pentacene:PS blend.** The addition of an insulating polymer such as PS to the organic ink has shown to be an effective way to improve thin-film processability and stability, and to boost the performance of TIPS-pentacene transistors[23,25]. Bearing this in mind, thin films based on different TIPS-pentacene:PS blending ratios, 4:1, 2:1, and 1:1, were deposited at a fixed coating speed of 10 mm/s in order to investigate its effect on the OFET performance and, in turn, on X-rays detection. Devices based on only TIPS-pentacene were used for comparison and, for simplicity, from now on we will refer to these devices as 1:0 films. In Supplementary Fig. 7, the polarized optical microscopy images of the 4:1, 2:1, and 1:1 deposited thin films show that TIPS-pentacene crystallizes in uniform thin films showing in all cases the spherulitic morphology previously observed for the 1:0 thin films. X-ray diffraction measurements (Supplementary Fig. 8) of the four blends exhibit identical diffraction patterns in agreement with the triclinic phase previously reported for this molecule[34], ensuring that the same crystal phase is present in all of them. Only sets of ($00l$) peaks were observed, indicative of the high crystallinity and orientation with respect to the substrate. The broader diffraction peaks registered for TIPS-pentacene:PS films as well as the small

shift in the 2-theta values are explained by the reduction of the crystalline domains size, as observed in the microscope images (see Supplementary Fig. 8 and Fig. 1), the lower thickness of the crystalline material layer and possible crystal strain effects that might cause the polymer binder[33]. The thin-films morphology was further characterized by AFM (see Supplementary Fig. 9), and the resulting surface roughness (*rms*) and thickness values for 1:0, 4:1, 2:1, and 1:1 blending ratios are reported in Supplementary Table 2. The thin-film thickness was between 40 and 90 nm. The smoothness of the PS-containing thin films is a proof of the uniformity provided by the polymer, which reduces roughness by almost one order of magnitude. This is an indication that TIPS-pentacene crystals are embedded in a polymeric matrix[22].

A statistical analysis of the main OFET parameters of 1:0, 4:1, 2:1, and 1:1 TIPS-pentacene:PS devices was performed employing different batches of samples to test reproducibility (ten devices for each formulation). Statistics on the saturation field-effect mobility and hole trap density ($N_T$) are depicted in Fig. 4a, b, while analysis of the on/off ratio ($I_{on}/I_{off}$) and threshold voltage ($V_{th}$) are collected in Supplementary Fig. 10. Comparison between all studied blends shows a general improvement of the electrical performance when PS is added to the formulation. In particular, it can be observed that the on/off ratio (Supplementary Fig. 10a) and the mobility (Fig. 4a) are higher for PS-containing devices. For 4:1 and 2:1 blends, the average values reach $4 \cdot 10^5$ and $0.5\,\mathrm{cm^2\,V^{-1}\,s^{-1}}$, respectively, which represents an increase of almost two orders of magnitude with respect to the pure TIPS-pentacene devices. Another benefit provided by the use of PS is found in the threshold voltage (Supplementary Fig. 10b), that shifts from positive values around 6 V for pure TIPS-pentacene devices to $V_{th}$ mean values between $-1$ and $-2\,\mathrm{V}$ in the blended films.

Interestingly, a steeper turn-on, and thus a lower SS, was found for PS-containing devices. This result indicates a smaller interfacial

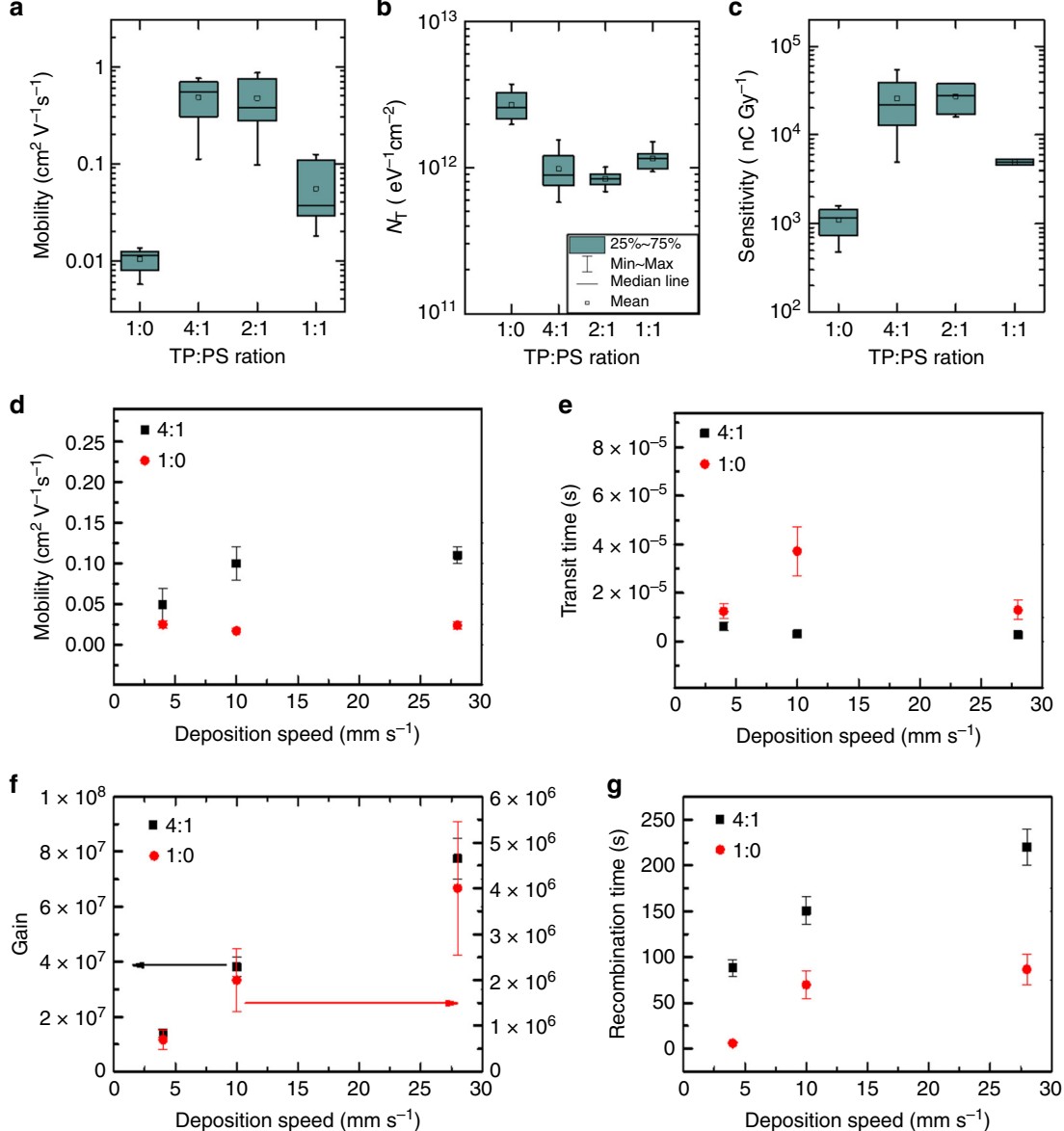

**Fig. 4 Impact of OFET electrical performance on X-ray detection sensitivity. a, b** Box-plot statistics for the OFET mobility and hole trap density, respectively, obtained from ten devices of each TIPS-pentacene:PS blend ratio (1:0, 4:1, 2:1, and 1:1). **c** Device X-ray sensitivity of 1:0, 4:1, 2:1, and 1:1 TIPS-pentacene: PS devices. The error bars refer to the statistical fluctuations of the parameters over ten samples for each blend ratio. Comparison between pure TIPS-pentacene (1:0) and blended (4:1) devices of: **d** experimentally determined mobility and **e** corresponding transit time; analytically determined **f** photoconductive gain and **g** recombination time, for different deposition speeds. The error bars refer to the statistical fluctuations of the parameters over four samples for each deposition speed.

hole trap density (Fig. 4b), which was estimated following Eq. (6) reported in the "Methods" section. Indeed, it has already been reported for various systems of organic semiconductor:PS thin films processed by solution that the crystalline semiconducting layer is formed on top of a PS buffering layer[25,35–37]. Thanks to this vertical phase separation, the PS layer in direct contact with the silicon dielectric plays an important role passivating the surface and reducing the density of hole traps at the dielectric/organic semiconductor interface, thus justifying the observed increased mobility.

The response to X-rays was tested for all the proposed TIPS-pentacene: PS formulations, 1:0, 4:1, 2:1, and 1:1. Figure 4c shows the estimated sensitivity values, which are higher for PS-containing devices, especially for 4:1 and 2:1 blends. Remarkably, an outstanding X-ray sensitivity as high as $1.3 \cdot 10^4 \, \mu C \, Gy^{-1} \, cm^{-2}$,

normalized to the pixel area, was obtained, which is the highest value reported for organic X-ray detectors. The sensitivity is about three and two orders of magnitude higher than the reference value for a-Se and poly-CZT X-ray detectors, respectively[28]. As a matter of fact, the gain mechanism in the blended devices is so efficient that the X-ray sensitivity values are comparable with direct detectors based on thick polycrystalline perovskite devices[38]. Noteworthy attention has to be paid in the comparison between the sensitivity values reported in the literature, since the units of measure and the types of normalization reported can be very different due to the plethora of organic materials/device geometries recently investigated for direct X-ray detection. In order to make a clear comparison between the sensitivity values presented in this work and the top sensitivities recently reported in the literature for organic[15,29], hybrid organic/inorganic[4,6,8], perovskite[38], and

**Table 2 Comparison between the SoA values of sensitivities and those reported in this work.**

| | Active layer | X-rays source | $S_A$ ($\mu C\ Gy^{-1}\ cm^{-2}$) | $S_v$ ($\mu C\ Gy^{-1}\ cm^{-3}$) |
|---|---|---|---|---|
| Kabir and Kasap[28] | Stabilized a-Se | 20 keV | 27 | |
| | Poly-CZT | | 291 | |
| Büchele et al.[4] | P3HT:PCBM + NP-GOS:Tb | W anode; 70 kVp; 2.5 mm Al filter | 7.3 | $7.3 \cdot 10^{3a}$ |
| Thirimanne et al.[6] | P3HT:PC$_{70}$BM + NP-Bi$_2$O$_3$ | W anode; 50 kVp; no filter | $3.9 \cdot 10^{3a}$ | $1.7 \cdot 10^6$ |
| Jayawardena et al.[8] | P3HT:PC$_{70}$BM + NP-Bi$_2$O$_3$ | W anode; 70 kVp; 2.5 mm Al filter | $2.8 \cdot 10^{3a}$ | $1.6 \cdot 10^5$ |
| Lai et al.[29] | TIPS-pentacene | Mo anode; 35 kVp; no filter | $4.8^a$ | $4.8 \cdot 10^{5a}$ |
| Ciavatti et al.[15] | TIPGe-pentacene | Mo anode; 35 kVp; no filter | $17.8^a$ | $8.9 \cdot 10^5$ |
| Liang et al.[39] | a-Ga$_2$O$_3$ | Cu anode; 40 kVp; no filter | 6.77 | $2.71 \cdot 10^5$ |
| Kim et al.[38] | MAPbI$_3$ | W anode; 100 kVp; 3 mm Al filter | $1.1 \cdot 10^4$ | $1.3 \cdot 10^{5a}$ |
| This work | TIPS-pentacene:PS | Mo anode; 35 kVp; no filter | $1.3 \cdot 10^4$ | $3.2 \cdot 10^9$ |

$^a$Values extracted by the device information reported in the referenced papers.

inorganic for large-area X-ray detectors[28,39], we summarize in Table 2 the sensitivity per unit area and per unit volume calculated using the same units of measurements.

The normalization per unit area has been calculated considering the pixel area (dashed red box in Fig. 2a), and not the active channel area, i.e., the product between the channel width and the channel length, to be fairer in the comparison between different geometries (e.g., vertical-stacked structures) with respect to those which use interdigitated electrodes, as in the present work.

In this view, the comparison of the sensitivity per unit area highlights the operative detector performance, whereas the sensitivity per unit volume highlights the material property, as it is typically reported for organic X-ray detectors. Among the results in the literature, Thirimanne et al.[6] reported a value of $1.7 \cdot 10^6\ \mu C\ Gy^{-1}\ cm^{-3}$ for hybrid organic inorganic devices with a vertical diode architecture, where a bulk heterojunction–nanoparticles (BHJ-NP) composite is sandwiched between indium tin oxide (ITO) and aluminum (Al) electrodes. More recently, Jayawardena et al.[8] employed a similar BHJ-NP photodiode architecture to demonstrate hole transport lengths up to 1 mm, enabling thick detectors with sensitivity of $1.6 \cdot 10^5\ \mu C\ Gy^{-1}\ cm^{-3}$. Our previous works[15,29] concerned flexible X-ray detectors with an OFET architecture employing an all organic active layer based on TIPS-pentacene and its derivatives. In those cases, we obtained sensitivities up to $17.8\ \mu C\ Gy^{-1}\ cm^{-2}$ per unit area and $8.9 \cdot 10^5\ \mu C\ Gy^{-1}\ cm^{-3}$ per unit volume. The corresponding sensitivity value per unit volume of the present work has the exceptional value of $3.2 \cdot 10^9\ \mu C\ Gy^{-1}\ cm^{-3}$, therefore it is three orders of magnitude higher than hybrid organic/inorganic photodiodes recently reported in the literature and four orders of magnitude higher than state-of-the-art all-organic X-ray detectors. The high X-ray sensitivity of these devices clearly points out their potential to real applications, although depending on the specific application other aspects such as long-term stability should be considered and technologically addressed.

Noteworthy, the observed trend of X-ray sensitivity versus TIPS-pentacene: PS ratio is very similar to the one shown by the OFET mobility (Fig. 4a). This correlation shows how the improvement of the transistor electrical performance, i.e., the hole mobility, due to the PS passivating action for hole traps at the semiconductor/dielectric interface, positively impacts on the detection performance, decreasing the holes transit time $\tau_t$ (Eq. 2). In other words, it indicates that improving the electrical performance of the transistor allows to optimize the device for the detection of high-energy radiation. However, as pointed out in the previous paragraph, the detection mechanism based on the photoconductive gain effect is assisted by the presence of the traps for minority carriers, i.e., electrons in this case, in the organic semiconductor layer. In Fig. 4 we report the experimentally determined mobility and the values of $\tau_t$, $\tau_r$, and gain $G$ calculated from the fit of the

experimental data for both 1:0 and 4:1 devices with the active layer deposited at different speeds. The plots clearly show how, for both pure TIPS-pentacene samples and blended ones, for similar mobility values, the variation of the photocurrent gain with increasing deposition speed follows the variation of the recombination time (affected by the electron traps).

Further, the transfer characteristics of the OFETs (TIPS-pentacene:PS 4:1 and pure TIPS 1:0) reveal an increase of the hysteresis with the deposition speed (Supplementary Fig. 11). This behavior is coherent with the increase of density of grain boundaries for high deposition speeds, which increases the number of electron traps in the film. In particular, the recent literature reported the correlation between the electron trap density in p-type OFETs and the grain size, i.e., the grain boundary density, of the organic semiconducting layer[40]. It is suggested that we could consider these traps as shallow traps, which can be easily charged or discharged during the sweep of the gate voltage. The anticlockwise direction of hysteresis in a p-type transistor suggests that such traps are traps for electrons.

In order to assess the reliability and stability of our devices, we investigated the shelf life of the best performing blends, 4:1 and 2:1, by monitoring the performance of the detectors for up to 80 days. During this period, the devices were stored and measured under dark and ambient conditions. Both the field-effect mobility and the X-ray sensitivity evolution are depicted in Supplementary Fig. 12. A decrease in the X-ray response of 30–40% was observed after more than 2 months from fabrication, as well as a device mobility drop of 45% in average. These changes are often attributed in the literature to degradation due to ambient humidity, an issue that is usually avoided by depositing an encapsulation layer on top of the organic film, that is not present in the here reported devices[41,42]. Indeed, TIPS-pentacene: PS devices show a more stable behavior than those based on pure TIPS-pentacene, which exhibit a strongly doped and degraded OFET performance after a few days from fabrication (Supplementary Fig. 13). These results demonstrate how TIPS-pentacene: PS blended films constitute an excellent and reliable material platform for the fabrication of long-lasting sensor devices due to the stability provided by the PS.

Up to now, we showed that by controlling the morphology and the electrical performance of TIPS-pentacene:PS devices, we obtained record sensitivities among organic direct X-ray detectors. However, together with sensitivity, the minimum detectable dose rate, i.e., the capability to detect very low doses, is a fundamental figure of merit that allows to fully assess the performance of X-ray detectors. Considering the high sensitivity values and the excellent electrical performance provided by TIPS-pentacene:PS devices, we performed a series of measurements to evaluate the photocurrent response of a 4:1 device at very low

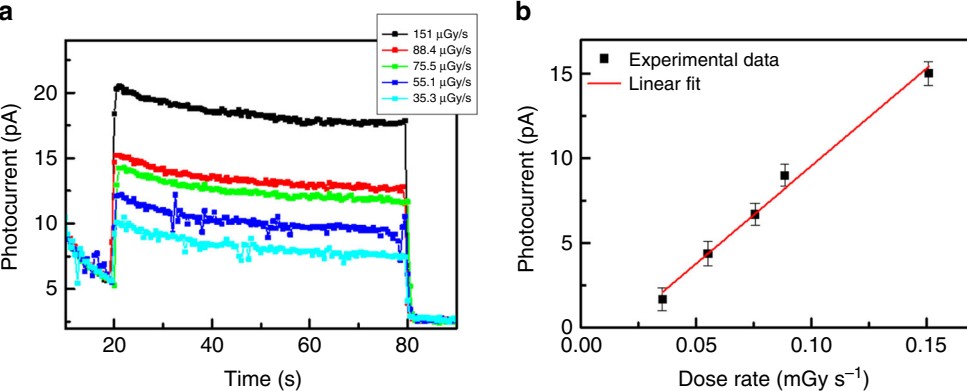

**Fig. 5 Detector response at low X-ray dose rates. a** X-ray induced photocurrent response of a TIPS-pentacene:PS 4:1 device at several dose rates, down to 35 µGy s$^{-1}$. **b** Photocurrent versus dose rate, including the linear fit used for the sensitivity calculation. The error bars refer to the statistical fluctuations of the signal amplitude over three on/off switching cycles of the X-ray beam in the same condition.

dose rates. To this aim, a system of Al filters was built to attenuate the dose rate provided by the X-ray tube, reaching values as low as a few µGy s$^{-1}$. Also, in order to minimize noise and light-induced photogenerated charges, the sample was kept in dark in a metal Faraday cage beforehand, which ensured a stable electrical behavior. In addition, the device was operated at $V_{SG} = 0$ V to decrease the dark current. In Fig. 5a, the photocurrent generated by the irradiation of the sample at several dose rates down to 35 µGy s$^{-1}$ is reported. Even at this faster time, response has to be ascribed to the operational regime used for these measurements (i.e., $V_{SG} = 0$ V), in which the charge carrier density in the transistor channel is low and the photocurrent results from the separation and collection of ionization charges generated by X-ray photon absorption[7]. As it can be seen in Fig. 5b, by operating at low dose rates we can obtain a sensitivity as high as $32 \pm 2$ µC Gy$^{-1}$ cm$^{-2}$, as calculated as the slope of the linear fit of the experimental data. This aspect is especially appealing for medical applications, where the intensity of the delivered radiation dose should be as low as possible. Typical dose rate values presently used in medical diagnostics are about 5.5 µGy s$^{-1}$ [43,44] and standard radiographic examinations have average effective total doses in the range 0.005–10 mGy[45].

Further, thanks to the unique feature of the here presented radiation sensors, we could obtain a very stable sensor based on a Wheatstone Bridge (see Supplementary Note 1 and Supplementary Fig. 14), where even small anomalies, such as the bias stress and the ageing effects, can be compensated and corrected by a fine tuning through the bridge architecture. By exploiting the similar electrical behavior of the four devices composing the bridge, we could obtain an overall stable and constant dark current that represents a highly desirable property to grant a reliable and reproducible response of radiation detectors.

## Discussion

To conclude, we investigate the origin of the physical processes and parameters controlling the minority carrier traps (electrons) that assist the photoconductive gain effect in X-ray direct radiation detectors based on TIPS-pentacene OFETs. We exploit the unique feature of BAMS technique, i.e., the tuning of the active layer's morphology and the OFETs mobility, as a tool to control and maximize the detection capability of such devices, both in terms of enhanced sensitivity and limit of detection. The results reported in this work lead to the following main conclusions. We demonstrate that by reducing the grain size and increasing the number of grain boundaries, we can increase the density of electron trap states within the material, enhancing the photoconductive gain for the

X-ray induced photocurrent. Further, by adding PS to the semiconductor solution, we reduce the interface hole trap density and consequently the charge carrier mobility is enhanced, as well as the device sensitivity.

By taking into account and controlling both factors, we reached record sensitivity values as high as $1.3 \cdot 10^4$ µC Gy$^{-1}$ cm$^{-2}$, the highest reported for organic-based direct X-ray detectors[6,15] and comparable with thick film perovskite ones[38], together with a very low minimum detectable dose rate, 35 µGy s$^{-1}$.

Thanks to the high reproducibility and uniformity provided by the BAMS deposition technique that allows to scale-up the fabrication to multipixels over large areas, we have implemented a proof-of-concept for a four-pixel detector in a Wheatstone bridge geometry, obtaining a very reliable sensor with a stable and low dark current, a highly desirable feature for radiation detectors.

This study gives an insight on the understanding of the crucial parameters and physical processes that control the X-rays detection performance of organic polycrystalline thin-film semiconductors, fundamental steps in order to implement real-life applications of direct high-energy radiation detection based on organic thin films.

## Methods

**Device fabrication**. Devices were fabricated onto Si/SiO$_2$ (200 nm SiO$_2$) substrates from Si-Mat with interdigitated Cr/Au electrodes, patterned by photolithography and deposited by thermal evaporation. The channel's width and length were 2.5 mm and 25 µm, respectively ($W/L = 100$). This geometry yields a pixel area of $4.25 \cdot 10^{-3}$ cm$^2$.

TIPS-pentacene and PS of molecular weight $M_w = 10,000$ g mol$^{-1}$ were purchased from Ossila and Sigma-Aldrich, respectively, and used without further purification. In the case of blend solutions, both components were separately dissolved in anhydrous chlorobenzene (2 wt%) and mixed at a volume ratio of 4:1, 2:1, or 1:1 (TIPS-pentacene:PS). Solutions based on pure TIPS-pentacene (ratio 1:0) were prepared at a 4 wt% concentration.

Before the thin-film deposition, substrates were cleaned by high-performance liquid chromatography quality acetone and isopropanol. The gold surface of the source and drain electrodes was chemically modified with a self-assembled monolayer of pentafluorobenzenethiol (PFBT, from Sigma-Aldrich). The substrates were first cleaned using ultraviolet ozone for 25 min, and then immersed into a $15 \cdot 10^{-3}$ M solution of PFBT in isopropanol for 15 min. Finally, the substrates were rinsed with pure isopropanol, and then dried under a nitrogen flux. Thin films were deposited at ambient conditions by the BAMS technique using a home-designed bar coater (see Supplementary Fig. 15 for more details on the technique). The substrates ($28 \times 16$ mm) were placed on the coating bed, kept at 105 °C, and about 40 µL of the blend solution was dispensed between the substrate and the bar. Immediately after that, the meniscus was sheared usually at speed of 10 mm s$^{-1}$ (lower and higher speeds were also tested). In all cases, the film dries immediately after the solution deposition.

**Thin-film characterization**. Optical microscope pictures were taken using an Olympus BX51 equipped with polarizer and analyzer.

X-ray diffraction measurements were carried out with a Siemens D-5000 diffractometer.

AFM images were obtained using a Park NX10 system using PPP-NCHR tips (Nanosensors) in noncontact mode and applying adaptive scan rate to slow down scan speed at crystallite borders. Subsequent data analysis was done using the Gwyddion software. In order to calculate the grain sizes, a dedicated Gwyddion tool has been used. In particular, setting a height threshold value, this tool can recognize different grains laying in the same thin film.

**Electrical characterization.** OFET electrical performance was measured on fresh devices using an Agilent B1500A semiconductor device analyzer connected to the samples with a Karl SÜSS probe station, at ambient conditions.

The field-effect mobility ($\mu$) and threshold voltage ($V_{th}$) were extracted in the saturation regime from a linear fit of the plot ($I_{SD}$)$^{1/2}$ versus $V_{SG}$. Mobility was extracted using the relationship

$$\mu = \frac{2L}{WC} \times \left( \frac{\partial \sqrt{|I_{SD}|}}{\partial V_{SG}} \right)^2, \tag{4}$$

where $C$ is the insulator capacitance per unit area, and $W$ and $L$ are the channel width and length, respectively.

The SS was calculated using the following equation:

$$SS = \left( \frac{\partial \log |I_{SD}|}{\partial V_{SG}} \right)^{-1}. \tag{5}$$

The interfacial trap density for the majority charge carriers (i.e., holes) per unit area ($N_T$) is directly proportional to SS, and has been estimated using the following equation:

$$N_T \approx \frac{C}{q^2} \left[ \frac{q \, SS}{k_B \, T \, \ln(10)} - 1 \right], \tag{6}$$

where $q$ is the electronic charge, $k_B$ is the Boltzmann constant, and $T$ is the absolute temperature.

During X-ray irradiation tests, the electrical photoresponse of the devices was measured by using a Keithley 2614 SourceMeter, controlled by a custom made Labview software. All measurements were carried out keeping the device in dark, in a Faraday cage, to reduce electrical noise and avoid light-induced photogeneration in the organic semiconductor.

The Wheatstone bridge was tested using a Keithley 6517A Electrometer in order to measure the $V_{out}$ signal. The power supply of the circuit ($V_{AB} = -20$ V) and the gate voltage applied to each device of the bridge ($V_{SG} = -5$ V) were imposed by a Keithley 2614 SourceMeter.

**X-ray irradiation.** Characterization under X-rays was performed using the X-ray broad spectrum provided by a molybdenum tube with an accelerating voltage of 35 kV and dose rates in the range 5–55 mGy s$^{-1}$, measured with an error below 5% by means of BARRACUDA X-Ray Analyzer from RTI.

For the low dose measurements, the samples were kept in dark in a metal Faraday cage for a long period of time (2 months) to minimize noise and the Vis-light-induced photogenerated current. A system of Al filters was built to allow working with X-ray doses as low as a few µGy s$^{-1}$.

## Data availability

The data that support the findings of this study are available from the corresponding author upon reasonable request.

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

## Acknowledgements

I.F., L.B., A.C., and B.F. acknowledge funding from INFN through the CSN5 FIRE project. This work was also funded by the ERC StG 2012-306826 e-GAMES and the DGI (Spain) project FANCY CTQ2016-80030-R. The authors also thank the Generalitat de Catalunya (2017-SGR-918), the Networking Research Center on Bioengineering, Bio-materials, and Nanomedicine (CIBER-BBN), and the Spanish Ministry of Economy and Competitiveness, through the "Severo Ochoa" Programme for Centers of Excellence in R&D (SEV-2015-0496). I.T. and A.T. are enrolled in the Materials Science PhD program of Universitat Autònoma de Barcelona and acknowledge FPU fellowship from the Spanish Ministry.

## Author contributions

L.B., A.C., and I.T. conceived and designed the experiments. I.T. and A.T. fabricated the devices and carried out electrical measurements of as-prepared OFET devices. L.B. performed the preliminary electrical measurements under the X-rays. A.C. built the experimental setup and implemented the software for data acquisition. I.T. and I.F. carried out the devices' characterization under X-rays and data analysis reported. I.F. and A.C. carried out fitting and simulation of experimental curves. I.T. and L.B. wrote the first draft of the manuscript. All authors discussed the results and revised the manuscript. B.F. and M.M.-T. coordinated the project.

## Competing interests

The authors declare no competing interests.
