## [Peer Review File · Nature Communications]

Reviewers' Comments:

Reviewer #1:

Remarks to the Author:

In the manuscript submitted by Temiño et al. they indicate that increasing the number of grain boundaries as a means of increasing the electron trap density, in combination with increasing the hole mobility through passivation of traps for holes, enables a high sensitivity to be recorded in their organic transistor X-ray detectors. In terms of the overall work discussed, it is the reviewer's opinion that while this work is of interest for those in the field, the audience is more specialist. Furthermore, there are a number of aspects (noted below) that need to be addressed prior to any further consideration for publication in any journal.

1) Figure 1d: It is advised that the authors identify and name the components shown in the photograph as the relevance of this image is somewhat limited at present to the general audience.

2) In Table 1, the thickness of the films are reported to increase with increasing deposition speed with ~50% increase in the film thickness at a coating speed of 28mm/s. The Gain and Sensitivity values reported in the same table appear to be correlated with this thickness increase while there does not appear to be a correlation between gain and mobility. The authors attribute this observation (in page 7) to an increase in the electron traps. It would be best if the authors can estimate the electron trap density to confirm this, and add additional proof to the manuscript. It appears to be more speculative at present.

3) In page 8, the authors indicate that the thin film X-ray diffraction measurements for different PS loadings are identical in nature. Looking at Figure S7, the diffraction pattern for the 1:0 appears to have sharper peaks compared to the PS incorporated samples. Furthermore, visually, it appears as if though the incorporation PS results in broadening of the peaks and possibly a shift in the 2-theta values. It is recommended that the authors study this in more detail as there is a possible relationship between the disorder and the X-ray sensitivity values in Figure 3 (c).

4) In page 4, the authors refer to equation (3) which is placed in the methods section. However, there is no indication in the main text where equation (3) is to be found. It is suggested that either the equations be moved into the main body or that the discussion indicates where the equations can be found.

5) In Figure 2e, the authors employ the X-ray induced photocurrent vs dose rate curve to calculate the sensitivity. However, the transient nature of the X-ray photocurrent should result in a time varying sensitivity which needs to be commented upon. Was there an integration time window applied? To clarify,

a) Which value of the photocurrent was used in calculating the sensitivity?

b) How does the sensitivity vary over time?

The authors should discuss both these aspects in detail, including possible dark current impacts.

6) In page 9, the authors indicate that in calculating the sensitivity, the pixel area was used. However, it is not entirely clear what values are considered in this definition. Is this the total area including that of the interdigitated fingers or is it only the interdigitated finger area? A schematic to clarify this would be helpful.

7) In Table 2, the authors have compared sensitivity numbers for several types of X-ray detectors based on area and volume scaling. However in the opinion of this reviewer, sensitivity is also dependant on the spectrum used (e.g. is there filtering, what is the tube kV etc.) Therefore, direct comparison of sensitivity is unlikely to be accurate. Therefore, the authors should add details regarding the spectra used by the different work compared into the table.

8) In Figure 4a, it appears as if though the transient response of the TIPS pentacene 4:1 device is

a lot faster than that reported in Figure 1e, f and also 5c (where a similar device was used). Typically it is expected that in the operational mode used here by the authors, the response would be a lot slower than that seen in Figure 4a. The authors should explain this potential discrepancy in characteristics.

Reviewer #2:

Remarks to the Author:

Overall, this is an excellent and well supported work. The authors have posed an interesting hypothesis, that the morphology drives the photoconductive gain of organic-semiconductor based X-ray sensors, and compellingly demonstrated that the higher trap density increases the effective responsibility primarily through an increase in the photoconductive gain of the devices in a series of three measurements. The morphology was also engineered using a blended host, and shown to further modify the performance in a manner consistent with the trap photoconductivity hypothesis.

While the sample set is limited in size, the experiments are sound, the results are well structured, and the conclusions are well supported by the data and analysis provided. I especially appreciate the comprehensive (and properly normalized) benchmarking provided by the authors to place the work into better context.

Some lifetime and physical measurement results are also presented.

I believe that this is suitable for publication in Nature Communications with the consideration of a few minor corrections.

Minor comments:

The response curves 1E and 1F should explain the bias conditions and configuration. The model used for photoconductive gain (even if just linear) should also be included, at least by reference, in the figure (it is covered in references 21 and 29, but these are not introduced until later).

I'm not sure that the Wheatstone bridge section is really necessary...using modern electronics direct measurements are as good as bridge measurements, and unless there is a foundational decrease in noise or other factor impacting the detectivity, it's not clear to me this section adds anything foundational (I'm OK with it staying in, but it would tighten the paper to skip it).

Reviewer #3:

Remarks to the Author:

This report concerns the manuscript "Morphology and mobility as tools to control and unprecedentedly enhance X-ray sensitivity in organic thin-film devices" by Temiño and co-workers. Their work focuses on the morphology of a polymer blend films with surprisingly large X-ray sensitivity. The manuscript is written in acceptable English; it is illustrated with high-quality figures. The experimental methods contain sufficient detail on most aspects, but since the contribution focuses on the effect of processing parameters on microstructure, it would be useful to include some more information on the home-built setup (including the geometry) and the drying lengths. It will be hard to compare the results to those obtained in coating setups in any case.

Flexible X-ray detectors based on organic and hybrid materials remain an active field of research. Many challenges remain in the field, although a lack of sensitivity is not really the main concern. The authors correctly mention a range of strategies that enhance the sensitivity; another important aspect (that they do not mention in the introduction) is the film thickness: it is possible to create very thick organic or hybrid layers if necessary. This can lead to a decrease of spatial resolution depending; such trade-offs are not discussed by the authors. This may not be necessary

here, but it indicates that one needs to be cautious when using sensitivity per volume as sole metric for material performance. I understand that the section on sensitivity in comparison to other materials was added in response to the critique that there was too little novelty; this is, however, a slippery slope. It is difficult to compare this polymer system with composite and hybrid detectors, and it is questionable whether the sensitivity alone is relevant given the practical challenges.

The manuscript focuses on TIPS-pentacene with its relatively low average Z and studies the mechanisms that lead to its sensitivity. I believe that the important question is whether insights on film formation truly provide new strategies towards, or an improved understanding of film properties. My report focuses on this question and not on the "practical relevance" of these films for potential commercial detectors.

OFET devices were used in order to measure charge carrier mobilities, and simple X-ray experiments were used in order to measure the induced photocurrent for the pure TIPS films. A photoconductive gain mechanism has been described for this material by the authors. Data on the effect of film thickness is lacking. The hypothesis that "a higher number of electron traps" causes greater amplification is not supported by independent data other than the model from a previous publication that is compared with the measurements provided here. What we do have is a correlation between grain size and sensitivity and the implicit assumption that all is as it was describe before (which does probably not merit an additional publication).

The study on the polymer-TIPS blends is more relevant and provides more detail: the reduced subthreshold swings point to traps indeed. The authors use it directly to calculate trap densities (Figure 3 b), but the subsequent logic is somewhat blurred. What is the relation between the majority carrier trap density and the X-ray sensitivity? Does the electron trap density stay constant? In their previous paper, the authors deemed the electron density relevant for the amplification mechanism. What is the mechanism here, and which role does the passivation by the polymer play? What is the novelty compared to the postulated mechanism, and how does the passivation work (if this is the critical aspect)?

I have mixed feelings on the comparison on page 9 and in table 2. Surely one can normalize all performance data in such a way that this material appears in a positive light, but does this add to understanding? The device sensitivity depends on the achievable thickness (not to speak of long-term stability, dark current and other factors) that are not compared here. I think it is not a bad idea to include this comparison – perhaps it triggers a debate – but it mainly indicates that something interesting is happening in these blends. The question is: what?

The last paragraph on page 10 is confusing. I do not understand what do the authors mean when they write "All in all, tuning either the thin film morphology and/or the OFET mobility have proven to be two independent and excellent strategies to enhance X-ray sensitivity." Surely mobility is (partially) a result of microstructure, and so is the enhancement mechanism (if it exists) that seem to be responsible for the high conversion rate. Where is the explanation for the extraordinary enhancement, though?

The authors proceed to demonstrate the functionality of their detectors at low doses and in a Wheatstone geometry. I have no problem with these results, but they lead away from the core question in my opinion.

In summary, this is an interesting contribution with one striking result but too little explanation of the observation. Perhaps it was necessary to include the comparison on page 9/10 in order to show how surprising the conversion efficiency reported here is. But given the large number of competing materials, device architectures, and possible relevant performance metrics, an explanation of the surprising increase should be the core piece of the manuscript. In its present state the contribution fails to deliver a coherent and convincing hypothesis that is supported by

data, or at least fails to present the existing data in such a way that a convincing picture emerges. Merely showing a surprisingly performance increase is probably not enough, in particular since the preparation of such materials is not always easy to reproduce in other laboratories; if the origin of the improvement remains unclear there is a considerable risk that others will not find it and have no way to understand why they do not.

I cannot, therefore, recommend acceptance of the contribution in its current form, but I think that the authors may be willing to invest some work into a better explanation. This would surely be more convincing than playing the old "highly improved property" game alone.

Reviewers' comments:

Reviewer #1 (Remarks to the Author):

In the manuscript submitted by Temiño et al. they indicate that increasing the number of grain boundaries as a means of increasing the electron trap density, in combination with increasing the hole mobility through passivation of traps for holes, enables a high sensitivity to be recorded in their organic transistor X-ray detectors. In terms of the overall work discussed, it is the reviewer's opinion that while this work is of interest for those in the field, the audience is more specialist. Furthermore, there are a number of aspects (noted below) that need to be addressed prior to any further consideration for publication in any journal.

1) *Figure 1d: It is advised that the authors identify and name the components shown in the photograph as the relevance of this image is somewhat limited at present to the general audience.*

1) In the revised manuscript we have modified Fig.1d, adding the description of the components showed in the picture to make them easily identifiable by a broader audience. We thank the reviewer for her/his suggestion, we report the modified Fig.1d hereafter for the Reviewer's convenience. In addition, in the new Figure S15 a closer photograph of the substrate has been included in order to better visualize the device layout.

2) *In Table 1, the thickness of the films are reported to increase with increasing deposition speed with ~50% increase in the film thickness at a coating speed of 28mm/s. The Gain and Sensitivity values reported in the same table appear to be correlated with this thickness increase while there does not appear to be a correlation between gain and mobility. The authors attribute this observation (in page 7) to an increase in the electron traps. It would be best if the authors can estimate the electron trap density to confirm this, and add additional proof to the manuscript. It appears to be more speculative at present.*

2) We thank the Reviewer for arising this issue, giving us the opportunity to better clarify the correlation between gain, sensitivity and film thickness in our OFET-based X-ray detectors. Indeed, referring to Table 1, samples with similar thickness (Low and Standard), exhibit different sensitivity and gain values. On the other hand, samples with very different thickness (Standard and High) show more similar values of sensitivity and gain. This disentanglement between the organic film thickness and the sensitivity to X-rays is not surprising. In fact, the here presented detectors are based on a field effect transistor structure where the charge transport occurs at the interface between the organic semiconductor and the dielectric layer, i.e. the few nanometers of the transistor channel. This differs from what happens in vertical stacked organic detectors, e.g. the one recently reported in literature by Jayawardena et al. [ACS Nano 2019, 13, 6973–6981], who presented a 100 μm thick P3HT:PCBM:Bi₂O₃ heterojunction device, where charge transport occurs through the bulk of the organic layer and thus its thickness plays a crucial role in the detection process. In our case the radiation induced additional charge carriers (holes) involved in the

photoconductive gain process flow within the OFET channel, meaning that the detection performance is more affected by the transport at the semiconductor/dielectric interface than by the absorption of the radiation in the semiconductor bulk. For this reason, increasing the active layer thickness does not represent a winning strategy to improve the X-ray detection properties of the here reported devices.

In the revised manuscript we have added a paragraph (page 9) to better clarify the thickness of the organic layer in the detection process.

We understand the Reviewer's concern about the lack of a direct experimental measurement of the electron trap density to confirm our results. However, the characterization of traps for minority charge carriers in organic semiconductors is challenging due to their longer relaxation times compared to conventional semiconductors. In fact, the exploitation of techniques able to distinguish between majority and minority carrier traps as Deep Level Transient Spectroscopy (DLTS) is not trivial: it was originally developed for inorganic materials and it requires enough high variations in the transient capacitance upon applying a voltage pulse, difficult to obtain with an organic material. However, even if such direct measurements are not viable, we have been able to estimate how the presence of the electron traps impacts on the X-ray detection process through the direct correlation between charge carrier recombination time and detection sensitivity.

In order to address the Reviewer's concern, we provide an additional analytical calculation of recombination time, i.e. the electron lifetime, for different deposition speeds of pure TIPS-pentacene based samples, that correspond to different morphologies of the organic film (i.e. different grain sizes).

We fitted the experimental data with the stretched exponential decay of the photocurrent after irradiation, given by the photoconductive gain model, which describes the process of amplification of the photocurrent generated by X-ray absorption in organic thin films, activated by the trapping of minority charge carriers (electrons in this case). The gain factor G can be expressed as the ratio between the recombination time (τ_r) and the holes transit time (τ_t):

$$G = \frac{\tau_r}{\tau_t}$$

We were thus able to assess the correlation of the experimentally measured sensitivity with the recombination time, which is related to the electron traps:

$$\tau_r(\rho_x) = \frac{\alpha}{\gamma} \left[\alpha \ln \left(\frac{\rho_0}{\rho_x} \right) \right]^{\frac{1-\gamma}{\gamma}}$$

where α is the time-scale in which the relaxation after the irradiation takes place, γ represents the width of the distribution of relaxation time-scale α_i (typically $\gamma < 1$ in amorphous and polycrystalline materials), ρ_0 is the carrier density in the saturation condition and ρ_x the carrier density induced by a certain dose of radiation.

They both increase when increasing the deposition speed, i.e. with the reduction of grain size and, as a consequence, with the increase of grain boundaries density.

On the other hand, we could assess that a different trend is observed with the experimentally measured hole mobilities, which is in line with the trend found for the related holes transit time (τ_t), calculated as follows:

$$\tau_t = \frac{L^2}{\mu V}$$

We report these plots below for the Reviewer's convenience.

Figure 3: Thin films of bare TIPS-pentacene. Experimentally determined (a) mobility and (b) sensitivity and analytical determined (c) hole transit time and (d) recombination time.

Moreover, we repeated the measurements under X-rays and the analytical calculations for samples deposited at different deposition speeds using TIPS-pentacene:PS blended organic semiconductor. As stated in the manuscript, the PS passivates the traps for majority carriers (holes) leading to an increase of the mobility, i.e. a reduction of the holes transit time along the channel and therefore an increase of the photocurrent gain in the blended devices. However, from the plots reported below (that have been included in Figure 4 of the revised manuscript) it is clear how, for both pure TIPS-pentacene samples and blended ones, for similar mobility values, the variation of the photocurrent gain with the increasing of the deposition speed follows the variation of the recombination time (affected by the electron traps).

Figure 4: comparison between pure TIPS-pentacene and blended TIPS-pentacene:PS thin films: (d) experimentally determined mobility and (e) corresponding transit time; analytical determined (f) photoconductive gain and (g) recombination time.

In the revised manuscript we have added some paragraph in the main text (page 5, 6, 8, 12) and Figure 3 and 4 to better clarify this issue.

Further, the transfer characteristics of the OFETs (TIPS-pentacene:PS 4:1 and bare TIPS-pentacene) reveals an increase of the hysteresis with the deposition speed. This behavior is coherent with the increase of density of grain boundaries for high deposition speeds, which increases the number of electron traps in the film. In particular, recent literature reported the correlation between the electron trap density in p-type OFETs and the grain size, i.e. the grain boundary density, of the organic semiconducting layer. [Gao et al.J. Mater. Chem. C, 2018, 6, 12498–12502]. It is suggested that we could consider these traps as shallow traps, which can be easily charged or discharged during the sweep of the gate voltage. The anticlockwise direction of hysteresis in a p-type transistor suggests that such traps are traps for electrons.

Figure S11: Transfer characteristics of (a) TIPS-pentacene:PS and (b) bare TIPS-pentacene 1:0 OFETs for low (4 mm/s) and high (28 mm/s) deposition speed.

In the revised manuscript we have added a paragraph main text (page 13) and Figure S11 in supplementary section to explain this point.

3) In page 8, the authors indicate that the thin film X-ray diffraction measurements for different PS loadings are identical in nature. Looking at Figure S7, the diffraction pattern for the 1:0 appears to have sharper peaks compared to the PS incorporated samples. Furthermore, visually, it appears as if though the incorporation PS results in broadening of the peaks and possibly a shift in the 2-theta values. It is recommended that the authors study this in more detail as there is a possible relationship between the disorder and the X-ray sensitivity values in Figure 3 (c).

We fully agree with the Reviewer with his/her comment and we apologise for the confusion.

It is true the fact that the diffraction peaks of blended samples exhibit broader peaks, often less intense and some small shifting is often observed. These effects are commonly observed and reported in the literature (see for instance: Adv. Sci. 2018, 5, 1700290, Adv. Funct. Mater. 2016, 26, 2379).

To understand this, it has to be first taken into account the different amount of crystalline material present. In films of only the OSC, we have tens of nm of crystalline material. However, it should be noticed that in the films based on blends, we only have a few monolayers of active material (see our work published at ACS Appl. Mater. Interfaces 2018, 10, 7296). Therefore, the amount of crystalline material is much smaller in the blended films and, thus, more interfacial effects are observed, whereas in the thicker bare OSC films the bulk material is dominating. Further, in the blended films, the crystallization of the OSC within the polymer matrix might lead to some strain in the crystal structure, as previously reported in other cases (Nature 2011, 480, 504–508). All these effects lead to less intense peaks, small shifts in the 2-theta values and also peak broadening.

Our main point in the manuscript was to report that all the films show the same crystal orientation and structure. However, to make it clearer we have modified the text as follows:

Page 10: “X-ray diffraction measurements (Fig. S7) of the four blends exhibit identical diffraction patterns in agreement with the triclinic phase previously reported for this molecule,³² ensuring that the same crystal phase is present in all of them. Only sets of (00l) peaks were observed, indicative of the high crystallinity and orientation with respect to the substrate. The broader diffraction peaks registered for TIPS-pentacene:PS films as well as the small shift in the 2-theta values are explained by the reduction of the crystalline domains size, as observed in the microscope images (see Fig.S6 and Fig. 1), the lower thickness of the crystalline material layer and possible crystal strain effects that might cause the polymer binder [Nature 2011, 480, 504–508].”

4) In page 4, the authors refer to equation (3) which is placed in the methods section. However, there is no indication in the main text where equation (3) is to be found. It is suggested that either the equations be moved into the main body or that the discussion indicates where the equations can be found.

4) We apologize for the inaccuracy. Following the Reviewer's suggestion, in the revised manuscript (page 10) we have added few words to explain where equation (3) (that has become equation (6)) can be found.

5) In Figure 2e, the authors employ the X-ray induced photocurrent vs dose rate curve to calculate the sensitivity. However, the transient nature of the X-ray photocurrent should result in a time varying sensitivity which needs to be commented upon. Was there an integration time window applied? To clarify,

a) Which value of the photocurrent was used in calculating the sensitivity?

b) How does the sensitivity vary over time? The authors should discuss both these aspects in detail, including possible dark current impacts.

5) The Reviewer comment is correct, we did not give details on the time window used to calculate the sensitivity. All the values reported in the manuscript have been calculated considering the X-ray induced photocurrent after 60 s of exposure to X-rays in order to have a reliable comparison of the detector performance in steady-state. We chose this time window because, as can be noted in the graph below, the variation of sensitivity over time increases with the exposure time reaching a saturation after about 50 s. However, it is noteworthy that the detector could be reliably used also for shorter exposure times, after a proper calibration, even if with a lower sensitivity to the radiation.

This exposure-time dependence of the sensitivity is in line with the analytical model used to fit the experimental data, as underlined by the figure below, reported in our previous publication [Basiricò, L. et al. Direct X-ray photoconversion in flexible organic thin film devices operated below 1 V. Nat. Commun. 7, 13063 (2016).], Figure 3c: for a given dose rate value, the sensitivity increases with the exposure time reaching a plateau:

In the revised manuscript we added the information about the photocurrent value used for the calculation of the sensitivity (page 5). Also, we added the plot of sensitivity variation over time in SI as new Figure S3.

6) In page 9, the authors indicate that in calculating the sensitivity, the pixel area was used. However, it is not entirely clear what values are considered in this definition. Is this the total area including that of the interdigitated fingers or is it only the interdigitated finger area? A schematic to clarify this would be helpful.

6) The pixel area used for the calculation of the sensitivity is the total area including the interdigitated electrodes. In accordance to the Reviewer's request and for clarity, in the revised manuscript we have added a dashed red box in the picture in Figure 2a indicating such area.

7) In Table 2, the authors have compared sensitivity numbers for several types of X-ray detectors based on area and volume scaling. However in the opinion of this reviewer, sensitivity is also dependant on the spectrum used (e.g. is there filtering, what is the tube kV etc.) Therefore, direct comparison of sensitivity is unlikely to be accurate. Therefore, the authors should add details regarding the spectra used by the different work compared into the table.

7) We agree with the Reviewer, the details of X-ray spectrum used are very important for an accurate comparison. Therefore, following the Reviewer's suggestion, in the revised version we included a column in Table 2 reporting the available details of the X-rays sources used in the different works. We also reported it below for the Reviewer's convenience.

8) In Figure 4a, it appears as if though the transient response of the TIPS pentacene 4:1 device is a lot faster than that reported in Figure 1e, f and also 5c (where a similar device was used). Typically, it is expected that in the operational mode used here by the authors, the response would be a lot slower than that seen in Figure 4a. The authors should explain this potential discrepancy in characteristics.

8) The reason of the different time scales can be ascribed to the different bias operation regime needed to assess the lowest detectable X-ray dose of the detector. In fact, the device was operated at $V_{SG} = 0$ V to decrease the dark current. In this operational regime the charge carrier density in the transistor channel is low and the detection process is no more ruled by the photoconductive gain effect, which is based on the balance between the holes flowing in the channel and the recombination with the trapped electrons, leading to the slow transient response reported in Figure 1e, f and 5c. In more detail, when $V_{SG} = 0$ is applied, the X-ray induced photocurrent in the organic semiconductor results from the charges photogenerated by radiation absorption, that are collected by the applied electric field between the source and the drain electrodes ($V_{DS} = -20$ V). Therefore, the response rise time is fast as it depends only on the timescale necessary to transport charges to the collecting electrode. The analysis of both these regimes has been carried out in a previous publication of our group [Ciavatti et al. Appl. Phys. Lett. 111, 183301 (2017)].

In the revised manuscript we modified the text to clarify this point (page 14). Also, we have modified Figure 4 (new Figure 5) showing the photocurrent curves and linear fit of the experimental data used to calculate the sensitivity. We report the modified figure below:

Reviewer #2 (Remarks to the Author):

Overall, this is an excellent and well supported work. The authors have posed an interesting hypothesis, that the morphology drives the photoconductive gain of organic-semiconductor based X-ray sensors, and compellingly demonstrated that the higher trap density increases the effective responsibility primarily through an increase in the photoconductive gain of the devices in a series of three measurements. The morphology was also engineered using a blended host, and shown to further modify the performance in a manner consistent with the trap photoconductivity hypothesis.

While the sample set is limited in size, the experiments are sound, the results are well structured, and the conclusions are well supported by the data and analysis provided. I especially appreciate the comprehensive (and properly normalized) benchmarking provided by the authors to place the work into better context.

Some lifetime and physical measurement results are also presented.

I believe that this is suitable for publication in Nature Communications with the consideration of a few minor corrections.

Minor comments:

The response curves 1E and 1F should explain the bias conditions and configuration. The model used for photoconductive gain (even if just linear) should also be included, at least by reference, in the figure (it is covered in references 21 and 29, but these are not introduced until later).

We thank the Reviewer for her/his accuracy. In the revised manuscript we have added the bias conditions in Fig.1e and 1f. Also, we have added the reference for the model applied for the fit, i.e. Ref.21, in the text (page5,6), in the legend of Fig.1f and in the caption.

I'm not sure that the Wheatstone bridge section is really necessary...using modern electronics direct measurements are as good as bridge measurements, and unless there is a foundational decrease in noise or other factor impacting the detectivity, it's not clear to me this section adds anything foundational (I'm OK with it staying in, but it would tighten the paper to skip it).

In the revised manuscript we have moved the paragraph to the SI section as supplementary note 2 and Fig.S14. We also removed the statement about this result in the abstract.

Reviewer #3 (Remarks to the Author):

This report concerns the manuscript “Morphology and mobility as tools to control and unprecedently enhance X-ray sensitivity in organic thin-film devices” by Temiño and co-workers. Their work focuses on the morphology of a polymer blend films with surprisingly large X-ray sensitivity. The manuscript is written in acceptable English; it is illustrated with high-quality figures. The experimental methods contain sufficient detail on most aspects, but since the contribution focuses on the effect of processing parameters on microstructure, it would be useful to include some more information on the home-built setup (including the geometry) and the drying lengths. It will be hard to compare the results to those obtained in coating setups in any case.

We thank the Reviewer for his/her comments and accordingly we included in the Supporting Information the layout of our deposition technique as well as some photos of our device layout (figure below). Further, in the experimental section we included more details regarding the substrates size and the drying time of the solutions, which in fact are dried immediately after deposition.

New Figure S15

Flexible X-ray detectors based on organic and hybrid materials remain an active field of research. Many challenges remain in the field, although a lack of sensitivity is not really the main concern. The authors correctly mention a range of strategies that enhance the sensitivity; another important aspect (that they do not mention in the introduction) is the film thickness: it is possible to create very thick organic or hybrid layers if necessary. This can lead to a decrease of spatial resolution depending; such trade-offs are not discussed by the authors. This may not be necessary here, but it indicates that one needs to be cautious when using sensitivity per volume as sole metric for material performance. I understand that the section on sensitivity in comparison to other materials was added in response to the critique that there was too little novelty; this is, however, a slippery slope. It is difficult to compare this polymer system with composite and hybrid detectors, and it is questionable whether the sensitivity alone is relevant given the practical challenges.

The manuscript focuses on TIPS-pentacene with its relatively low average Z and studies the mechanisms that lead to its sensitivity. I believe that the important question is whether insights on film formation truly

provide new strategies towards, or an improved understanding of film properties. My report focuses on this question and not on the “practical relevance” of these films for potential commercial detectors.

OFET devices were used in order to measure charge carrier mobilities, and simple X-ray experiments were used in order to measure the induced photocurrent for the pure TIPS films. A photoconductive gain mechanism has been described for this material by the authors. Data on the effect of film thickness is lacking. The hypothesis that “a higher number of electron traps” causes greater amplification is not supported by independent data other than the model from a previous publication that is compared with the measurements provided here. What we do have is a correlation between grain size and sensitivity and the implicit assumption that all is as it was describe before (which does probably not merit an additional publication).

- We thank the Reviewer for arising, in line with one of the comments of Reviewer #1, the issue of the lack of discussion about the effect of film thickness on the detection performance in the here reported OFET-based X-ray detectors. Indeed, referring to Table 1, samples with similar thickness (Low and Standard), exhibit different sensitivity and gain values. On the other hand, samples with very different thickness (Standard and High) show similar values of sensitivity. This disentanglement between the organic film thickness and the sensitivity to X-rays is not surprising. In fact, the here presented detectors are based on a field effect transistor structure where the charge transport occurs at the interface between the organic semiconductor and the dielectric layer, i.e. the few nanometers of the transistor channel. This differs from what happens in vertical stacked organic detectors, e.g. the one recently reported in literature by Jayawardena et al. [ACS Nano 2019, 13, 6973–6981], who presented a 100 μm thick P3HT:PCBM:Bi2O3 heterojunction device, where charge transport occurs through the bulk of the organic layer and thus its thickness plays a crucial role in the detection process. In our case the radiation induced additional charge carriers (holes) involved in the photoconductive gain process flow within the OFET channel, meaning that the detection performance is more affected by the transport at the semiconductor/dielectric interface than by the absorption of the radiation in the semiconductor bulk. For this reason, increasing the active layer thickness does not represent a winning strategy to improve the X-ray detection properties of the here reported devices

In the revised manuscript we have added a short paragraph (page 9) to better clarify the thickness of the organic layer in the detection process.

In order to address the Reviewer’s request of major support on the correlation between the electron traps at the grain boundaries, which density can be tuned through the deposition speed, and the sensitivity to X-rays, we provide an additional analytical calculation of recombination time, i.e. the electron lifetime, for different deposition speeds of pure TIPS-pentacene based samples, that correspond to different morphologies of the organic film (i.e. different grain sizes).

We fitted the experimental data with the stretched exponential decay of the photocurrent after irradiation, given by the photoconductive gain model, which describes the process of amplification of the photocurrent generated by X-ray absorption in organic thin films, activated by the trapping of minority charge carriers (electrons in this case). The gain factor G can be expressed as the ratio between the recombination time (τ_R) and the holes transit time (τ_t):

$$G = \frac{\tau_r}{\tau_t}$$

We were thus able to assess the correlation of the experimentally measured sensitivity with the recombination time, which is related to the electron traps:

$$\tau_r(\rho_x) = \frac{\alpha}{\gamma} \left[\alpha \ln \left(\frac{\rho_0}{\rho_x} \right) \right]^{\frac{1-\gamma}{\gamma}}$$

where α is the time-scale in which the relaxation after the irradiation takes place, γ represents the width of the distribution of relaxation time-scale α_i (typically $\gamma < 1$ in amorphous and polycrystalline materials), ρ_0 is the carrier density in the saturation condition and ρ_x the carrier density induced by a certain dose of radiation.

They both increase when increasing the deposition speed, i.e. with the reduction of grain size and, as a consequence, with the increase of grain boundaries density.

On the other hand, we could assess that a different trend is observed with the experimentally measured hole mobilities, which is in line with the trend found for the related holes transit time (τ_t), calculated as follows:

$$\tau_t = \frac{L^2}{\mu V}$$

We report these plots below for the Reviewer's convenience.

Figure 3: Thin films of bare TIPS-pentacene. Experimentally determined (a) mobility and (b) sensitivity and analytical determined (c) hole transit time and (d) recombination time.

In the revised manuscript we have added a paragraph in the main text (page 5,6,8) and Figure 3 to better clarify this issue.

The study on the polymer-TIPS blends is more relevant and provides more detail: the reduced subthreshold swings point to traps indeed. The authors use it directly to calculate trap densities (Figure 3 b), but the subsequent logic is somewhat blurred. What is the relation between the majority carrier trap density and the X-ray sensitivity? Does the electron trap density stay constant? In their previous paper, the authors deemed the electron density relevant for the amplification mechanism. What is the mechanism here, and which role does the passivation by the polymer play? What is the novelty compared to the postulated mechanism, and how does the passivation work (if this is the critical aspect)?

Regarding the TIPS-pentacene:PS blended devices, in order to better clarify the issues arisen by the Reviewer, i.e. the relation between the majority carrier trap density and the X-ray sensitivity, the role of the electron trap density and that of the polymer, we repeated the measurements under X-rays and the analytical calculations for samples deposited at different deposition speeds. As stated in the manuscript, the PS passivates the traps for majority carriers (holes) leading to an increase of the mobility in the blended devices, i.e. a reduction of the holes transit time along the channel and therefore an increase of the photocurrent gain. However, from the plots reported below (that have been included in Figure 4 of the revised manuscript) it is clear how, for both pure TIPS-pentacene samples and blended ones, for similar mobility values, the increasing of the photocurrent gain with the increasing of the deposition speed (related to morphology and in particular to the density of grain boundaries) follows the variation of the recombination time (affected by the electron traps).

Figure 4: comparison between pure TIPS-pentacene and blended TIPS-pentacene:PS thin films: (d) experimentally determined mobility and (e) corresponding transit time; analytical determined (f) photoconductive gain and (g) recombination time.

In the revised manuscript we have added a paragraph in the main text (page 12) and Figure 4 to better clarify this issue.

Further, the transfer characteristics of the OFETs (TIPS-pentacene:PS 4:1 and bare TIPS-pentacene) reveals an increase of the hysteresis with the deposition speed. This behavior is coherent with the increase of density of grain boundaries for high deposition speeds, which increases the number of electron traps in the film. In particular, recent literature reported the correlation between the electron trap density in p-type OFETs and the grain size, i.e. the grain boundary density, of the organic semiconducting layer. [Gao et al. J. Mater. Chem. C, 2018, 6, 12498–12502]. It is suggested that we could consider these traps as shallow traps, which can be easily charged or discharged during the sweep of the gate voltage. The anticlockwise direction of hysteresis in a p-type transistor suggests that such traps are traps for electrons.

Figure S11: Transfer characteristics of (a) TIPS-pentacene:PS and (b) bare TIPS-pentacene 1:0 OFETs for low (4 mm/s) and high (28 mm/s) deposition speed.

In the revised manuscript we have added a paragraph main text (page 13) and Figure S11 in supplementary section to explain this point.

I have mixed feelings on the comparison on page 9 and in table 2. Surely one can normalize all performance data in such a way that this material appears in a positive light, but does this add to understanding? The device sensitivity depends on the achievable thickness (not to speak of long-term stability, dark current and other factors) that are not compared here. I think it is not a bad idea to include this comparison – perhaps it triggers a debate – but it mainly indicates that something interesting is happening in these blends. The question is: what?

We agree with the reviewer that for real applications the sensitivity is not the only parameter that should be taken into account and other aspects such stability, cost, amount of required material, etc. should be taken into consideration. However, here our goal was to compare this important figure of merit with commercial and reported state-of-the-art devices to demonstrate the potential of these materials, although there are other technological aspects very important that should be evaluated in order to transfer these materials to the market. At the end, the choice of materials/devices will also depend on the requirements of each specific application. Accordingly, in the text (page 11) we added the following statement:

“The high X-ray sensitivity of these devices clearly points out their potential to real applications, although depending on the specific application other aspects such as long-term stability should be considered and technologically addressed.”

The last paragraph on page 10 is confusing. I do not understand what do the authors mean when they write “All in all, tuning either the thin film morphology and/or the OFET mobility have proven to be two independent and excellent strategies to enhance X-ray sensitivity.” Surely mobility is (partially) a result of microstructure, and so is the enhancement mechanism (if it exists) that seem to be responsible for the high conversion rate. Where is the explanation for the extraordinary enhancement, though?

In the revised manuscript the paragraph has been substituted with a more extended explanation of the results, including the new Figures 4d, 4e, 4f, 4g. The enhancement of the X-ray detection performance in the blended devices is given, from the one side, by the passivation of the holes trap at the semiconductor/dielectric interface, which lead to the increase of the holes mobility, resulting in the decrease of holes transit time and, consequently, to an enhancement of the photocurrent gain and sensitivity. From the other side, we have demonstrated how, by tuning the deposition speed of the organic semiconductor, we are able to increase the grain boundaries and the associated electron traps, which assist

the detection process by increasing the recombination time and thus the gain of the X-ray induced photocurrent.

We report below again the formulas of the gain, the transit time and the recombination time for the Reviewer's convenience:

$$G = \frac{\tau_r}{\tau_t}$$

$$\tau_r(\rho_x) = \frac{\alpha}{\gamma} \left[\alpha \ln \left(\frac{\rho_0}{\rho_x} \right) \right]^{\frac{1-\gamma}{\gamma}}$$

$$\tau_t = \frac{L^2}{\mu V}$$

The authors proceed to demonstrate the functionality of their detectors at low doses and in a Wheatstone geometry. I have no problem with these results, but they lead away from the core question in my opinion.

In the revised manuscript we have moved the paragraph on the Wheatstone bridge to the SI section as supplementary note 2 and Fig.S14. We also removed the statement about this result in the abstract.

In summary, this is an interesting contribution with one striking result but too little explanation of the observation. Perhaps it was necessary to include the comparison on page 9/10 in order to show how surprising the conversion efficiency reported here is. But given the large number of competing materials, device architectures, and possible relevant performance metrics, an explanation of the surprising increase should be the core piece of the manuscript. In its present state the contribution fails to deliver a coherent and convincing hypothesis that is supported by data, or at least fails to present the existing data in such a way that a convincing picture emerges. Merely showing a surprisingly performance increase is probably not enough, in particular since the preparation of such materials is not always easy to reproduce in other laboratories; if the origin of the improvement remains unclear there is a considerable risk that others will not find it and have no way to understand why they do not.

I cannot, therefore, recommend acceptance of the contribution in its current form, but I think that the authors may be willing to invest some work into a better explanation. This would surely be more convincing than playing the old "highly improved property" game alone.

We thank the Reviewer for his/her comments, which were very useful to us in focusing our efforts to give a clearer and more effective explanation of the phenomenon. We hope that with the responses that we are providing and the changes we made in the revised manuscript, we convinced the Reviewer that our results go beyond the merely improvement of the performance. In particular, in our opinion this work gives a novel and deeper insight into the X-rays detection process in organic thin films, identifying the traps assisting the photocurrent gain process and proposing actual strategies to control it.

Reviewers' Comments:

Reviewer #1:

Remarks to the Author:

In the revised manuscript submitted, the authors have made satisfactory improvements to the presentation of the data, clarification of the technical questions raised and also have clarified the calculation aspects (definition of sensitivity, comparison of sensitivity results). I am happy to recommend the acceptance of this manuscript now for publication.

Reviewer #2:

None

Reviewer #3:

Remarks to the Author:

The authors, Beatrice Fraboni and colleagues, have submitted a revised version of their manuscript "Morphology and mobility as tools to control and unprecedentedly enhance X-ray sensitivity in organic thin-film devices" together with a detailed reply letter. One of the reviewers insinuated that the topic may be too specialized for this journal – I do not quite agree, in particular when considering film thickness and its effect on performance. This question seems of rather basic interest when discussing such sensors.

Two reviewers asked for clarification regarding the role of device thickness in detection. The authors now explain why thickness is much less relevant in their device architecture than in previously published work. They extended the part explaining how the microstructure in the film affects performance and clarified the link between carrier dynamics and detection performance.

A second, related concern was electron trap density: the manuscript highlighted its importance but did not provide direct data. The authors now added additional indirect measurements (obtained by changing the deposition speed during fabrication) to corroborate their hypotheses on the role of traps. I think that the results discussed in the added text on page 12 and 13 are very interesting and relevant. It is probably difficult to draw quantitative conclusions (the morphology variations with deposition speed likely comprise more than just grain size), but the trends are consistent with the proposed amplification mechanism.

In summary, I believe that the authors have addressed the concerns raised by the reviewers and the manuscript can be accepted for publication in Nature Communications.